# Factors Determining the Average Price Level: A Combined Microeconomic and Macroeconomic Approach

**Tamara Peneva Todorova** *  **and Brikena Myftarallari**

Department of Economics, American University in Bulgaria, 2700 Blagoevgrad, Bulgaria;
kenabusiness@gmail.com
* Correspondence: ttodorova@aubg.edu

**Abstract:** We analyze various determinants of the average price level using a strictly mathematical approach. Starting with the microeconomic perspective, we review the effect of demand shifters such as consumer income and the level of advertising on the average price level in a simple partial market equilibrium model. Then, we discuss the effect of supply shifters such as the exogenous tax level, worker wage, rental rate, and technology. We use implicit differentiation and Jacobian determinants. While government spending triggers inflation, taxes have the opposite effect. This is consistent with Keynesian theory. Money supply increases national income and prices while reducing the equilibrium interest rate. Therefore, money supply has pro-inflationary effects. The effect of money demand is the opposite—it increases the equilibrium interest rate, thereby lowering national income and prices. Augmenting the model to the level of international trade, we find that exports raise national income, the interest rate, and the average price level, while the effect of imports is just the opposite. Government spending raises the exchange rate while continuous inflation lowers it.

**Keywords:** inflation; government spending; taxes; exports multiplier

## 1. Introduction

One of the most essential debates in macroeconomics is about the origins of inflation. While the Keynesian school supports fiscal policy and increased government spending to stabilize the economy, the monetarist school recommends careful control over money supply and active monetary policy measures to prevent inflation. John Maynard Keynes prescribes government spending in times of a recession to give an impetus to the economy, but the monetarists led by Milton Friedman consider expansionary fiscal policy to have inflationary effects. They maintain that raising money supply increases only nominal national income without affecting the real one.

Keynes (1936) sees the increase in aggregate demand as the reason for demand-pull inflation. Since aggregate demand comprises consumption, investment, and government spending along the national income model, any factors behind these components drive aggregate demand and can cause demand-pull inflation. Inflation arises only when aggregate demand exceeds aggregate supply at the full employment level, and the larger the gap between the two, the bigger the rise in the price level. In "The General Theory of Employment, Interest, and Money", Keynes (1936) suggested that lowering any component of aggregate demand can alleviate the pressure on prices. While government spending may trigger inflation, tax increases can be used alone or in combination with other measures to manage money flow and reduce demand while containing inflation. In times of war when there is hyperinflation and when controlling the amount of money or reducing overall spending may not be feasible, an increase in tax can help control demand (Keynes 1936).

Monetarists object to this view, claiming that it is primarily money supply that causes inflation, and that inflation is always a purely monetary phenomenon which results when the quantity of money in circulation grows more quickly than output (Friedman 1970).

Money supply plays a key role in the economy according to the monetarists. Therefore, it should be controlled strictly through the means of monetary policy. Money supply is a key determinant of both output and prices in the short run and solely of prices in the long run (Friedman and Schwartz 1963). The ideological differences between the two schools lead to different policy prescriptions about the role of government in managing the economy. While Keynesians tolerate inflation to achieve social goals such as increased employment, monetarists prioritize price stability by curbing inflation.

While our analysis confirms Keynesian findings about the influence of certain macroeconomic variables on prices, we do not reject the monetarist view on the role of money supply. We demonstrate that government spending increases the average price level while taxes lower it. But it is equally true that increased money supply raises the price level, while the effect of money demand is the opposite. Thus, we do not see the Keynesian and the monetarist views as necessarily opposing. While the components of aggregate demand put an inflationary pressure on the economy, money supply increases nominal output, the average price level, and the equilibrium interest rate. The two theories, the Keynesian and the monetarist, complement each other according to our findings.

There are alternative theories which explain the rise of inflation. While Keynes emphasizes demand-pull inflation, the "new inflation" stream analyzes cost-push inflation. This type of inflation is nurtured by wage increases enforced by trade unions and higher profits demanded by businesses (Totonchi 2011). Phillips's article "The Relationship between Unemployment and the Rate of Change of Money Wage Rates in the United Kingdom, 1861–1957" is instrumental in this respect. Phillips (1958) observes that inflation arises when the growth rate of wages exceeds that of the productivity of labor. On the other hand, monopolies and oligopolies demanding higher profits raise prices above the normal level, which causes profit-push or price-push inflation. Cost-push inflation indicates that market power can be a source of inflation.

The structural theory of inflation maintains that inflation can be caused by structural imbalances in the economy, especially in underdeveloped countries. More specifically, distortions and bottlenecks in some sectors and the insufficient supply of food are at the root of inflation in countries in Latin America or in India (Prebisch 1950, 1961; Cardoso 1981). More recent studies on the structural perspective to inflation are provided by Taylor (2004), Barbosa-Filho and Taylor (2006), Barbosa-Filho (2014), and Kim (2023). While the increases in money supply and monetary policy are seen as the primary cause of inflation in developed nations, inflation is not solely a monetary problem in developing nations. The rational expectations school of economic thought takes issue with inflation, too. Factors generally associated with fiscal imbalances, such as greater money growth and currency rate depreciation resulting from a balance of payments crisis, dominate the inflation process in developing nations (Hansen and Sargent 1980; Sargent and Wallace 1981; Montiel 1989).[1] People make economic decisions based on their past experience and use all the information available to them in economic decision making. Those decisions are based on rationality, all the available information, and past experiences. Rational expectations explain anticipated inflation, which nurtures future inflation (Sargent 2013).

Our purpose is to study the effects of various macroeconomic variables on the average price level, pursuing a microeconomic and a macroeconomic approach. To achieve this goal, we use simple mathematical tools such as implicit differentiation, comparative-static derivatives, and Jacobian determinants. Using a simple framework within the national income model, we find that government spending increases national income, the equilibrium interest rate, and the price level. Adding taxes to the model demonstrates that they have the opposite effect on the macroeconomic variables in question and reduce the inflationary pressure on prices. We then add the money market to the model to account for money supply and money demand. We find that money supply increases national income and the price level while lowering the equilibrium interest rate. Money demand does the opposite, that is, it reduces national income and the price level while raising the interest rate.

We then expand the model by the volume of international trade. Consistent with Keynesian theory, exogenous exports increase national income, the interest rate, and the price level, thus, raising the standard of living in the country. Imports, on the other hand, do the opposite. As imports from abroad grow, national income, the interest rate, and domestic prices all diminish. We also find the government expenditure multiplier in the conditions of an open economy. As predicted by Keynes, both the exports multiplier and the government expenditure multiplier are positive.

This paper is organized as follows: Section 2 takes a microeconomic perspective to the equilibrium price of one good, that is, using a partial market equilibrium analysis. Section 3 reveals the national income model in the conditions of a closed economy. Starting from a basic two-equation model, we add exogenous taxes, the money market, and exogenous price, reflecting the initial price level. In Section 4, we expand the model to account for net exports and the volume of trade. The effect of government spending is the same both in a closed and an open economy. While exogenous exports tend to reduce domestic prices, exogenous imports have the opposite effect. The paper ends with concluding remarks in Section 5.

## 2. The Microeconomic Perspective: A Market Equilibrium Model

On a market for a single commodity, we have quantity demanded $q_d$ dependent on the own price of the good $p$ and exogenous consumer income $Y_o$. Supply, on the other hand, is given by $q_s$ and depends on the price of the good $p$ and the tax rate, which is exogenously determined. Thus, in equilibrium we have

$$q_d = D(p, Y_o) \qquad D_p < 0 \qquad D_{Y_0} > 0 \tag{1}$$

$$q_s = S(p, T_o) \qquad S_p > 0 \qquad S_{T_0} < 0 \tag{2}$$

Following the law of demand, quantity demanded is negatively related to price $p$ and positively to consumer income $Y_o$. At the same time, due to the law of supply, quantity supplied is positively related to the own price of the good but negatively to the average tax level $T_o$. We write the equilibrium condition as a single equation.

$$F = D(p, Y_o) - S(p, T_o) = 0 \tag{3}$$

Using the implicit function rule, we find the following comparative static derivatives:

$$\frac{d\overline{p}}{dY_o} = -\frac{F_{Y_o}}{F_p} = -\frac{D_{Y_o}}{D_p - S_p} = \frac{D_{Y_o}}{S_p - D_p} > 0 \tag{4}$$

$$\frac{d\overline{p}}{dT_o} = -\frac{F_{T_o}}{F_p} = \frac{S_{T_o}}{D_p - S_p} > 0, \tag{5}$$

where $F_p \neq 0$ since $F_p = D_p - S_p < 0$. Thus, equilibrium price is positively related to both exogenous consumer income and the tax rate. We can find the effect on equilibrium quantity indirectly through the supply function. While equilibrium quantity increases with an increase in consumer income for a normal good, it falls with an increase in the tax rate, as shown in Figure 1. An increase in consumer income for a normal good shifts the demand curve to the right (from $D$ to $D'$), with supply being the same. This increases both equilibrium quantity (from $q_1$ to $q_2$) and equilibrium price (from $p_1$ to $p_2$).

We can augment the model by adding more exogenous variables which affect the price level. For instance, on the demand side we can add the level of advertising, which is exogenously determined and shapes consumer tastes and preferences. Thus, advertising is a demand shifter similar to consumer income.

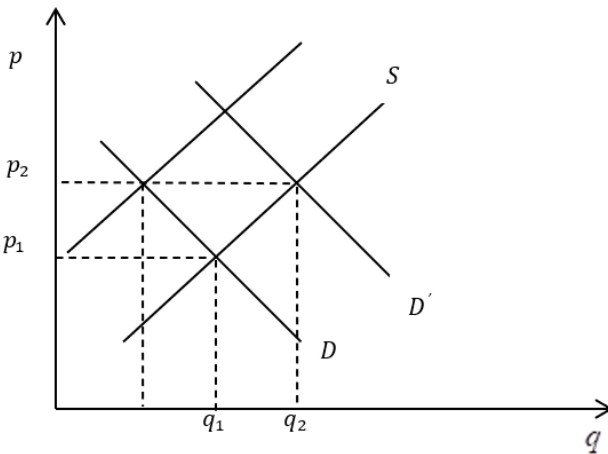

**Figure 1.** Comparative-static effect of an increase in consumer income and tax rate.

Some supply shifters that can be incorporated in the model, in addition to exogenous tax rate, are the wage rate, the rental rate, and the level of technology the firm is faced with. All these are assumed to be exogenously determined. Thus, the expanded market equilibrium model becomes as follows:

$$q_d = D(p, Y_o, A_o) \qquad D_p < 0 \qquad D_{Y_0} > 0 \qquad D_{A_0} > 0 \tag{6}$$

$$q_s = S(p, T_o, w_o, r_o, t_o) \qquad S_p > 0 \qquad S_{T_0} < 0 \qquad S_{w_0} < 0 \qquad S_{r_0} < 0 \qquad S_{t_0} > 0 \tag{7}$$

All comparative static derivatives exist and are known to us. It can be noted that on the demand side, advertising positively affects quantity demanded. By nurturing consumer tastes and preferences, a bigger advertising budget on the part of the firm stimulates consumer demand. On the supply side, an increase in equilibrium worker wage in the labor market or rental rate in the capital market, reflecting comparative-static changes in factor markets, affects supply negatively. As input prices increase, the quantity supplied by the individual firm decreases. The effect of technology on the quantity supplied is positive; that is, improvements in technology expand the production function and, as a result, the firm can supply more on the market. Applying the implicit function rule again, we check that $F_p \neq 0$ where $F_p = D_p - S_p < 0$. Furthermore, we obtain the following comparative static effects on price. Like consumer income, advertising is a demand shifter which increases equilibrium price.

$$\frac{d\overline{p}}{dA_o} = -\frac{F_{A_o}}{F_p} = -\frac{D_{A_o}}{D_p - S_p} = \frac{D_{A_o}}{S_p - D_p} > 0 \tag{8}$$

We previously found

$$\frac{d\overline{p}}{dY_o} = -\frac{F_{Y_o}}{F_p} = -\frac{D_{Y_o}}{D_p - S_p} = \frac{D_{Y_o}}{S_p - D_p} > 0, \tag{9}$$

and

$$\frac{d\overline{p}}{dT_o} = -\frac{F_{T_o}}{F_p} = \frac{S_{T_o}}{D_p - S_p} > 0, \tag{10}$$

On the supply side, exogenous wage and rental rate increase equilibrium price due to the fall in supply; that is,

$$\frac{d\overline{p}}{dw_o} = -\frac{F_{w_o}}{F_p} = \frac{S_{w_o}}{D_p - S_p} > 0, \tag{11}$$

and

$$\frac{d\overline{p}}{dr_o} = -\frac{F_{r_o}}{F_p} = \frac{S_{r_o}}{D_p - S_p} > 0 \tag{12}$$

Finally, by increasing supply at any level of the market price an improved technology leads to a fall in equilibrium price.

$$\frac{d\overline{p}}{dt_o} = -\frac{F_{t_o}}{F_p} = \frac{S_{t_o}}{D_p - S_p} < 0 \tag{13}$$

These results confirm economic theory, i.e., that increased input prices and the tax level shift the supply curve leftward, which causes a higher equilibrium price and a lower equilibrium quantity for a given demand curve. Technology, on the contrary, shifts supply to the right, which decreases equilibrium price and increases equilibrium quantity. Thus, the effect of technology on the price of finished goods is favorable.

## 3. The Macroeconomic Perspective: A National Income Model

Let us assume a simple national income model with average price level $p$ introduced.

$$Y = C(Y, p) + I(Y) + G_o \qquad 0 < C_Y = \frac{dC}{dY} < 1 \qquad C_p = \frac{dC}{dp} < 0 \tag{14}$$

$$p = p_o + g(Y) \qquad I_Y = \frac{dI}{dY} > 0 \qquad g_Y > 0 \tag{15}$$

where prices are a rising function of national income. There is an initial price level $p_o$ which is positively related to the average price level, and which affects national income as well. All functions are differentiable in all variables, so we can apply the implicit function theorem. To do that, we rewrite the system in an implicit form.

$$Y - C(Y, p) - I(Y) - G_o = 0 \tag{16}$$

$$p - p_o - g(Y) = 0 \tag{17}$$

Applying the implicit function theorem, we find the effect of exogenous government spending on the endogenous variables $Y$ and $p$.

$$\begin{bmatrix} 1 - C_Y - I_Y & -C_p \\ -g_Y & 1 \end{bmatrix} \begin{bmatrix} \frac{d\overline{Y}}{dG_o} \\ \frac{d\overline{p}}{dG_o} \end{bmatrix} = \begin{bmatrix} 1 \\ 0 \end{bmatrix} \tag{18}$$

$$|J| = 1 - C_Y - I_Y - g_Y C_p > 0 \tag{19}$$

We can expect the parenthesized expression in the Jacobian to be positive since $C_Y$ is the marginal propensity to consume and $I_Y$ should equal the marginal propensity to save in equilibrium. The sum of the two marginal propensities should not exceed 1. In equilibrium and in the absence of foreign trade, i.e., in the absence of the propensity to import, the determinant would be as follows:

$$|J| = -g_Y C_p > 0 \tag{20}$$

$$\frac{d\overline{Y}}{dG_o} = \frac{\begin{vmatrix} 1 & -C_p \\ 0 & 1 \end{vmatrix}}{|J|} = \frac{1}{1 - C_Y - I_Y - g_Y C_p} > 0 \tag{21}$$

As can be expected, the government expenditure multiplier is positive. This multiplier is higher the lower the effect of an increase in aggregate demand on prices and the less sensitive consumption is to price changes. The multiplier is also positively related to the two marginal propensities. If aggregate investment is assumed to be exogenous and the price equation is ignored, the government expenditure multiplier is exactly equal to the

investment multiplier and takes the standard value $\frac{d\overline{Y}}{dG_o} = \frac{1}{1-C_Y} > 0$. As to the effect of government spending on prices,

$$\frac{d\overline{p}}{dG_o} = \frac{\begin{vmatrix} 1 - C_Y - I_Y & 1 \\ -g_Y & 0 \end{vmatrix}}{|J|} = \frac{g_Y}{1 - C_Y - I_Y - g_Y C_p} > 0 \tag{22}$$

Government spending raises prices directly by increasing aggregate demand. The effect of government spending on price can also be found as a chain derivative by differentiating the second equation.

$$\frac{d\overline{p}}{dG_o} = \frac{d\overline{p}}{dY}\frac{d\overline{Y}}{dG_o} = \frac{g_Y}{1 - C_Y - I_Y - g_Y C_p} > 0 \tag{23}$$

Through the multiplier process, government spending raises aggregate income, which further results in an increase in the equilibrium price level. Hence, government spending seems to be a trigger for inflation. To find the effect of the base price level $p_o$ on the endogenous variables,

$$\begin{bmatrix} 1 - C_Y - I_Y & -C_p \\ -g_Y & 1 \end{bmatrix} \begin{bmatrix} \frac{d\overline{Y}}{dp_o} \\ \frac{d\overline{p}}{dp_o} \end{bmatrix} = \begin{bmatrix} 0 \\ 1 \end{bmatrix} \tag{24}$$

$$\frac{d\overline{Y}}{dp_o} = \frac{\begin{vmatrix} 0 & -C_p \\ 1 & 1 \end{vmatrix}}{|J|} = \frac{C_p}{1 - C_Y - I_Y - g_Y C_p} < 0 \tag{25}$$

$$\frac{d\overline{p}}{d\overline{p}_o} = \frac{\begin{vmatrix} 1 - C_Y - I_Y & 0 \\ -g_Y & 1 \end{vmatrix}}{|J|} = \frac{1 - C_Y - I_Y}{1 - C_Y - I_Y - g_Y C_p} > 0 \tag{26}$$

Inflation, which can be identified with the initial price level, affects national income negatively by reducing aggregate consumption. Economic growth is hampered by the level of inflation, with countries facing inflation being less likely to experience substantive economic growth. At the same time, the effect on equilibrium price is positive, as can be deduced from the second equation. This implies that inflation is self-nurtured. The initial level of inflation nurtures future inflation; that is, inflation seems to be persistent. Introducing taxes in the original national income model,

$$Y = C(Y_d, p) + I(Y) + G_o \qquad 0 < C_{Yd} < 1 \qquad C_p = \frac{dC}{dp} < 0 \tag{27}$$

$$Y_d = Y - T_o \qquad I_Y = \frac{dI}{dY} > 0 \tag{28}$$

$$p = p_o + g(Y) \qquad g_Y > 0 \tag{29}$$

Total tax collection $T_o$ is exogenous and represents the difference between total national income and disposable income $Y_d$. All derivatives exist while the propensity to consume $C_{Yd}$ is now the share of consumption in disposable, rather than total, income. Rewriting in an implicit form and applying the implicit function theorem again,

$$Y - C[Y_d(Y, T_o), p] - I(Y) - G_o = 0 \tag{30}$$

$$p - p_o - g(Y) = 0 \tag{31}$$

This gives rise to the following matrix equation.

$$\begin{bmatrix} 1 - C_{Yd} - I_Y & -C_p \\ -g_Y & 1 \end{bmatrix} \begin{bmatrix} \frac{d\overline{Y}}{dT_o} \\ \frac{d\overline{p}}{dT_o} \end{bmatrix} = \begin{bmatrix} -C_{Yd} \\ 0 \end{bmatrix} \tag{32}$$

$$|J| = 1 - C_{Yd} - I_Y - g_Y C_p > 0 \tag{33}$$

We again expect the parenthesized expression in the Jacobian to be positive since $C_{Yd}$ is the marginal propensity to consume and $I_Y$ is the marginal propensity to save in equilibrium. Thus, the sum of the two propensities cannot exceed 1.

$$\frac{d\overline{Y}}{dT_o} = \frac{\begin{vmatrix} -C_{Yd} & -C_p \\ 0 & 1 \end{vmatrix}}{|J|} = -\frac{C_{Yd}}{1 - C_{Yd} - I_Y - g_Y C_p} < 0 \tag{34}$$

$$\frac{d\overline{p}}{dT_o} = \frac{\begin{vmatrix} 1 - C_{Yd} - I_Y & -C_{Yd} \\ -g_Y & 0 \end{vmatrix}}{|J|} = -\frac{g_Y C_{Yd}}{1 - C_{Yd} - I_Y - g_Y C_p} < 0 \tag{35}$$

The effect of exogenous taxes $T_o$ is just opposite to that of government spending—since taxes reduce national income, aggregate demand falls, alleviating the inflationary pressure on prices. Therefore, taxes would be a good instrument in dealing with persistent inflation. This seems to be in accordance with the Keynesian theory of government spending and taxes as instruments in curbing inflation. The results confirm that while government spending fosters inflation, taxes have the opposite effect.

$$\frac{d\overline{p}}{dG_o} = \frac{d\overline{p}}{dY}\frac{d\overline{Y}}{dG_o} = \frac{g_Y}{1 - C_Y - I_Y - g_Y C_p} > 0 \tag{36}$$

A result we obtained previously is that through the multiplier process, government spending raises aggregate income, which further leads to an increase in equilibrium prices. Alternatively, we can account for the money market in the original national income model. We assume equilibrium in the money market, as represented by the equality of nominal money supply and the liquidity function. Liquidity (money demand) furthermore is a function of national income and the interest rate. It is positively related to national income $Y$ (transactional demand for money) but negatively to the equilibrium interest rate $i$ (speculative demand for money).

$$Y = C(Y,p) + I(Y,i) + G_o \quad 0 < C_Y = \frac{dC}{dY} < 1 \quad C_p = \frac{dC}{dp} < 0 \quad I_Y = \frac{dI}{dY} > 0 \quad I' = \frac{dI}{di} < 0 \tag{37}$$

$$\frac{M_{so}}{p} = L(Y,i) \qquad L_Y > 0 \qquad L_i < 0 \tag{38}$$

$$p = p_o + g(Y) \quad g_Y > 0 \tag{39}$$

Rewriting in an implicit form,

$$Y - C(Y,p) - I(Y,i) - G_o = 0 \tag{40}$$

$$L(Y,i) - \frac{M_{so}}{p} = 0 \tag{41}$$

$$p - p_o - g(Y) = 0, \tag{42}$$

or

$$\begin{bmatrix} 1 - C_Y - I_Y & -I' & -C_p \\ L_Y & L_i & \frac{M_{so}}{p^2} \\ -g_Y & 0 & 1 \end{bmatrix} \begin{bmatrix} \frac{d\overline{Y}}{dG_o} \\ \frac{di}{dG_o} \\ \frac{d\overline{p}}{dG_o} \end{bmatrix} = \begin{bmatrix} 1 \\ 0 \\ 0 \end{bmatrix} \tag{43}$$

$$|J| = I'\left(L_Y + g_Y \frac{M_{so}}{p^2}\right) + L_i(1 - C_Y - I_Y - g_Y C_p) < 0 \tag{44}$$

$$\frac{d\overline{Y}}{dG_o} = \frac{\begin{vmatrix} 1 & -I' & -C_p \\ 0 & L_i & \frac{M_{so}}{p^2} \\ 0 & 0 & 1 \end{vmatrix}}{|J|} = \frac{L_i}{I\prime(L_Y + g_Y \frac{M_{so}}{p^2}) + L_i(1 - C_Y - I_Y - g_Y C_p)} > 0 \tag{45}$$

$$\frac{d\overline{i}}{dG_o} = \frac{\begin{vmatrix} 1 - C_Y - I_Y & 1 & -C_p \\ L_Y & 0 & \frac{M_{so}}{p^2} \\ -g_Y & 0 & 1 \end{vmatrix}}{|J|} = -\frac{L_Y + g_Y \frac{M_{so}}{p^2}}{I\prime(L_Y + g_Y \frac{M_{so}}{p^2}) + L_i(1 - C_Y - I_Y - g_Y C_p)} > 0 \tag{46}$$

$$\frac{d\overline{p}}{dG_o} = \frac{\begin{vmatrix} 1 - C_Y - I_Y & -I' & 1 \\ L_Y & L_i & 0 \\ -g_Y & 0 & 0 \end{vmatrix}}{|J|} = \frac{L_i g_Y}{I\prime(L_Y + g_Y \frac{M_{so}}{p^2}) + L_i(1 - C_Y - I_Y - g_Y C_p)} > 0 \tag{47}$$

Government spending again stimulates national income, which further increases prices. Government spending also raises prices directly as part of aggregate demand. The result is very similar to the one in the absence of the money market equation; that is,

$$\frac{d\overline{p}}{dG_o} = \frac{g_Y}{1 - C_Y - I_Y - g_Y C_p} > 0 \tag{48}$$

At the same time, government spending pushes up the equilibrium interest rate, thus crowding out private investment. To analyze this effect more deeply, we transform the second derivative.

$$\frac{d\overline{i}}{dG_o} = -\frac{L_Y + g_Y \frac{M_{so}}{p^2}}{I'\left(L_Y + g_Y \frac{M_{so}}{p^2}\right) + L_i(1 - C_Y - I_Y - g_Y C_p)} = -\frac{\left(L_Y + g_Y \frac{M_{so}}{p^2}\right)}{L_i} \frac{d\overline{Y}}{dG_o} > 0 \tag{49}$$

We see that some of the variables that enforce the crowding out effect are the sensitivity of money demand to increases in the national income, of prices to increases in national income, money supply and the government-expenditure multiplier. Crowding out is lower the more responsive money demand is to changes in the interest rate and the higher the equilibrium price level. Furthermore, the less responsive investment is to changes in the interest rate; that is, the smaller $I'$, the smaller the denominator $|J|$ for the three comparative static derivatives. This means that government spending increases national income more, which further increases the interest rate and prices. To find the comparative static effect of money supply on the endogenous variables,

$$\begin{bmatrix} 1 - C_Y - I_Y & -I' & -C_p \\ L_Y & L_i & \frac{M_{so}}{p^2} \\ -g_Y & 0 & 1 \end{bmatrix} \begin{bmatrix} \frac{d\overline{Y}}{dM_{so}} \\ \frac{d\overline{i}}{dM_{so}} \\ \frac{d\overline{p}}{dM_{so}} \end{bmatrix} = \begin{bmatrix} 0 \\ \frac{1}{p} \\ 0 \end{bmatrix} \tag{50}$$

$$|J| = I'\left(L_Y + g_Y \frac{M_{so}}{p^2}\right) + L_i(1 - C_Y - I_Y - g_Y C_p) < 0 \tag{51}$$

$$\frac{d\overline{Y}}{dM_{so}} = \frac{\begin{vmatrix} 0 & -I' & -C_p \\ \frac{1}{p} & L_i & \frac{M_{so}}{p^2} \\ 0 & 0 & 1 \end{vmatrix}}{|J|} = \frac{I'}{pI\prime(L_Y + g_Y \frac{M_{so}}{p^2}) + pL_i(1 - C_Y - I_Y - g_Y C_p)} > 0 \tag{52}$$

$$\frac{d\bar{i}}{dM_{so}} = \frac{\begin{vmatrix} 1 - C_Y - I_Y & 0 & -C_p \\ L_Y & \frac{1}{p} & \frac{M_{so}}{p^2} \\ -g_Y & 0 & 1 \end{vmatrix}}{|J|} = \frac{1 - C_Y - \overset{+}{I_Y} - g_Y C_p}{pI\prime(L_Y + g_Y \frac{M_{so}}{p^2}) + pL_i(1 - C_Y - I_Y - g_Y C_p)} < 0 \quad (53)$$

$$\frac{d\bar{p}}{dM_{so}} = \frac{\begin{vmatrix} 1 - C_Y - I_Y & -I' & 0 \\ L_Y & L_i & \frac{1}{p} \\ -g_Y & 0 & 0 \end{vmatrix}}{|J|} = \frac{I' g_Y}{pI\prime(L_Y + g_Y \frac{M_{so}}{p^2}) + pL_i(1 - C_Y - I_Y - g_Y C_p)} > 0 \quad (54)$$

Consistent with Keynesian macroeconomic theory, money supply increases national income and reduces the equilibrium interest rate. Its effect on prices is positive since by increasing money supply the government stimulates national income and aggregate demand, which leads to higher equilibrium prices. A secondary mechanism by which money supply increases the price level is through investment, where money supply reduces the equilibrium interest rate which fosters investment. We now combine our equations in a comprehensive model accounting for both tax collection and the money market. More specifically, we introduce taxes as the difference between total and disposable income, on the one hand, and the equilibrium on the money market, on the other. With taxes introduced, consumption is again a function of disposable income $Y_d$. The model then becomes as follows:

$$Y = C(Y_d, p) + I(Y, i) + G_o \qquad 0 < C_{Yd} = \frac{dC}{dY_d} < 1 \qquad C_p = \frac{dC}{dp} < 0 \quad (55)$$

$$Y_d = Y - T_o \qquad I_Y = \frac{dI}{dY} > 0 \qquad I' = \frac{dI}{di} < 0 \quad (56)$$

$$\frac{M_{so}}{p} = L(Y, i) \qquad L_Y > 0 \qquad L_i < 0 \quad (57)$$

$$p = p_o + g(Y) \qquad g_Y > 0 \quad (58)$$

Rewriting the model implicitly,

$$-C[Y_d(Y, T_o), p] - I(Y, i) - G_o = 0 \quad (59)$$

$$L(Y, i) - \frac{M_{so}}{p} = 0 \quad (60)$$

$$p - p_o - g(Y) = 0 \quad (61)$$

For the effect of exogenous taxes, we obtain the following:

$$\begin{bmatrix} 1 - C_{Yd} - I_Y & -I' & -C_p \\ L_Y & L_i & \frac{M_{so}}{p^2} \\ -g_Y & 0 & 1 \end{bmatrix} \begin{bmatrix} \frac{d\bar{Y}}{dT_o} \\ \frac{d\bar{i}}{dT_o} \\ \frac{d\bar{p}}{dT_o} \end{bmatrix} = \begin{bmatrix} -C_{Yd} \\ 0 \\ 0 \end{bmatrix} \quad (62)$$

$$|J| = I'\left(L_Y + g_Y \frac{M_{so}}{p^2}\right) + L_i(1 - C_{Yd} - I_Y - g_Y C_p) < 0 \quad (63)$$

$$\frac{d\bar{Y}}{dT_o} = \frac{\begin{vmatrix} -C_{Yd} & -I' & -C_p \\ 0 & L_i & \frac{M_{so}}{p^2} \\ 0 & 0 & 1 \end{vmatrix}}{|J|} = -\frac{C_{Yd} L_i}{I\prime(L_Y + g_Y \frac{M_{so}}{p^2}) + L_i(1 - C_{Yd} - I_Y - g_Y C_p)} < 0 \quad (64)$$

$$\frac{d\bar{i}}{dT_o} = \frac{\begin{vmatrix} 1 - C_Y - I_Y & -C_{Yd} & -C_p \\ L_Y & 0 & \frac{M_{so}}{p^2} \\ -g_Y & 0 & 1 \end{vmatrix}}{|J|} = \frac{\overset{+}{C_{Yd}\left(L_Y + g_Y\frac{M_{so}}{p^2}\right)}}{I'(L_Y + g_Y\frac{M_{so}}{p^2}) + L_i(1 - C_{Yd} - I_Y - g_YC_p)} < 0 \quad (65)$$

$$\frac{d\bar{p}}{dT_o} = \frac{\begin{vmatrix} 1 - C_Y - I_Y & -I' & -C_{Yd} \\ L_Y & L_i & 0 \\ -g_Y & 0 & 0 \end{vmatrix}}{|J|} = -\frac{C_{Yd}g_YL_i}{I'(L_Y + g_Y\frac{M_{so}}{p^2}) + L_i(1 - C_{Yd} - I_Y - g_YC_p)} < 0 \quad (66)$$

Taxes reduce equilibrium national income and suppress aggregate demand, which results in lower prices. We obtain that taxes can serve as an anti-inflationary measure. Furthermore, as income decreases, transaction demand for money falls too, reducing the equilibrium interest rate. It is easy to see that these results conform to our previous ones in the absence of a money market equation.

$$\frac{d\bar{Y}}{dT_o} = -\frac{C_{Yd}}{1 - C_{Yd} - I_Y - g_YC_p} < 0$$

from Equation (34)

$$\frac{d\bar{p}}{dT_o} = -\frac{g_YC_{Yd}}{1 - C_{Yd} - I_Y - g_YC_p} < 0$$

from Equation (35).

We compare the derivatives $\frac{d\bar{Y}}{dT_o}$ and $\frac{d\bar{p}}{dT_o}$ with and without the money market considered. In the second case, the derivatives can be transformed as follows.

$$\frac{d\bar{Y}}{dT_o} = -\frac{C_{Yd}L_i}{I'\left(L_Y + g_Y\frac{M_{so}}{p^2}\right) + L_i(1 - C_{Yd} - I_Y - g_YC_p)} = -\frac{C_{Yd}}{\frac{I'}{L_i}\left(L_Y + g_Y\frac{M_{so}}{p^2}\right) + (1 - C_{Yd} - I_Y - g_YC_p)} < 0 \quad (67)$$

where the first term in the denominator disappears in the absence of a money market equation. Thus, the result is consistent with $\frac{d\bar{Y}}{dT_o}$ in the simple national income model. We observe the same with $\frac{d\bar{p}}{dT_o}$; that is,

$$\frac{d\bar{p}}{dT_o} = -\frac{C_{Yd}g_YL_i}{I'\left(L_Y + g_Y\frac{M_{so}}{p^2}\right) + L_i(1 - C_{Yd} - I_Y - g_YC_p)} = -\frac{C_{Yd}g_Y}{\frac{I'}{L_i}\left(L_Y + g_Y\frac{M_{so}}{p^2}\right) + (1 - C_{Yd} - I_Y - g_YC_p)} < 0 \quad (68)$$

which is $\frac{d\bar{p}}{dT_o} = -\frac{g_YC_{Yd}}{(1 - C_{Yd} - I_Y) - g_YC_p} < 0$ if the first term in the denominator is dropped, that is, with the money market ignored. For the government expenditure multiplier in the tax-augmented model, we have

$$\begin{bmatrix} 1 - C_{Yd} - I_Y & -I' & -C_p \\ L_Y & L_i & \frac{M_{so}}{p^2} \\ -g_Y & 0 & 1 \end{bmatrix} \begin{bmatrix} \frac{d\bar{Y}}{dG_o} \\ \frac{d\bar{i}}{dG_o} \\ \frac{d\bar{p}}{dG_o} \end{bmatrix} = \begin{bmatrix} 1 \\ 0 \\ 0 \end{bmatrix} \quad (69)$$

$$|J| = I'\left(\underset{-}{L_Y + g_Y\frac{M_{so}}{p^2}}\right) + L_i(\underset{-}{1 - C_{Yd} - I_Y - g_YC_p}) < 0 \quad (70)$$

$$\frac{d\bar{Y}}{dG_o} = \frac{\begin{vmatrix} 1 & -I' & -C_p \\ 0 & L_i & \frac{M_{so}}{p^2} \\ 0 & 0 & 1 \end{vmatrix}}{|J|} = \frac{L_i}{I'(L_Y + g_Y\frac{M_{so}}{p^2}) + L_i(1 - C_{Yd} - I_Y - g_YC_p)} > 0 \quad (71)$$

$$\frac{d\bar{i}}{dG_o} = \frac{\begin{vmatrix} 1 - C_Y - I_Y & 1 & -C_p \\ L_Y & 0 & \frac{M_{so}}{p^2} \\ -g_Y & 0 & 1 \end{vmatrix}}{|J|} = -\frac{\overset{+}{L_Y + g_Y \frac{M_{so}}{p^2}}}{I'\left(L_Y + g_Y \frac{M_{so}}{p^2}\right) + L_i(1 - C_{Yd} - I_Y - g_Y C_p)} > 0 \tag{72}$$

$$\frac{d\bar{p}}{dG_o} = \frac{\begin{vmatrix} 1 - C_Y - I_Y & -I' & 1 \\ L_Y & L_i & 0 \\ -g_Y & 0 & 0 \end{vmatrix}}{|J|} = \frac{g_Y L_i}{I'(L_Y + g_Y \frac{M_{so}}{p^2}) + L_i(1 - C_{Yd} - I_Y - g_Y C_p)} > 0 \tag{73}$$

These results are very similar to the model in the absence of taxation; that is,

$$\frac{d\overline{Y}}{dG_o} = \frac{L_i}{I'(L_Y + g_Y \frac{M_{so}}{p^2}) + L_i(1 - C_Y - I_Y - g_Y C_p)} > 0,$$

from Equation (45).

$$\frac{d\bar{i}}{dG_o} = -\frac{L_Y + g_Y \frac{M_{so}}{p^2}}{I'(L_Y + g_Y \frac{M_{so}}{p^2}) + L_i(1 - C_Y - I_Y - g_Y C_p)} > 0,$$

from Equation (46)

$$\frac{d\bar{p}}{dG_o} = \frac{L_i g_Y}{I'(L_Y + g_Y \frac{M_{so}}{p^2}) + L_i(1 - C_Y - I_Y - g_Y C_p)} > 0,$$

and Equation (47).

The difference is the marginal propensity to consume, which is computed based on total income in the first case and disposal income in the second case. Comparing $\frac{d\overline{Y}}{dT_o}$ with $\frac{d\overline{Y}}{G_o}$, we obtain

$$\frac{d\overline{Y}}{dT_o} = -\frac{C_{Yd} L_i}{I'\left(L_Y + g_Y \frac{M_{so}}{p^2}\right) + L_i(1 - C_{Yd} - I_Y - g_Y C_p)} = -C_{Yd} \frac{d\overline{Y}}{dG_o} < 0 \tag{74}$$

Since the marginal propensity to consume $C_{Yd}$ is less than 1, the decrease in national income due to taxes is smaller than the increase in it from government spending. Thus, government expenditure is justified according to the model. A simpler version of the model is one without a price equation while keeping money supply exogenous.

$$Y = C(Y) + I(Y, i) + G_o \qquad 0 < C_Y = \frac{dC}{dY} < 1 \qquad I_Y = \frac{dI}{dY} > 0 \qquad I' = \frac{dI}{di} < 0 \tag{75}$$

$$\frac{M_{so}}{p_o} = L(Y, i) \qquad L_Y > 0 \qquad L_i < 0 \tag{76}$$

We can find the effect of the exogenous price level $p_o$ on the endogenous variables and how good inflation is for the economy. Rewriting in an implicit form,

$$Y - C(Y) - I(Y, i) - G_o = 0 \tag{77}$$

$$\frac{M_{so}}{p_o} - L(Y, i) = 0 \tag{78}$$

$$\begin{bmatrix} 1 - C_Y - I_Y & -I' \\ -L_Y & -L_i \end{bmatrix} \begin{bmatrix} \frac{d\overline{Y}}{dp_o} \\ \frac{d\bar{i}}{dp_o} \end{bmatrix} = \begin{bmatrix} 0 \\ \frac{M_{so}}{p_o^2} \end{bmatrix} \tag{79}$$

$$|J| = -L_i(1 - C_Y - I_Y) - L_Y I' > 0 \tag{80}$$

$$\frac{dY}{dp_o} = \frac{\begin{vmatrix} 0 & -I' \\ \frac{M_{so}}{p_o^2} & L_i \end{vmatrix}}{|J|} = -\frac{I'M_{so}}{p_o^2[L_i(1 - C_Y - I_Y) + L_Y I']} < 0 \tag{81}$$

$$\frac{di}{dp_o} = \frac{\begin{vmatrix} 1 - C_Y - I_Y & 0 \\ -L_Y & \frac{M_{so}}{p_o^2} \end{vmatrix}}{|J|} = -\frac{(1 - C_Y - I_Y)M_{so}}{p_o^2[L_i(1 - C_Y - I_Y) + L_Y I']} > 0 \tag{82}$$

When the average price level increases, national income falls. To defeat inflation, the government must reduce money supply, but this affects national income negatively. Since money supply is reduced, the equilibrium interest rate increases on the money market. The same results are obtained with a version of the model where money demand is assumed to be exogenous while money supply depends positively on the equilibrium interest rate. Money supply is the prerogative of the central bank, but when the interest rate is too high, the central bank may decide to increase the quantity of money in circulation. In case the interest rate is too low, the central bank may opt to reduce money supply to normalize the interest rate. Thus, for the derivative $M'_s = \frac{dM_s}{di}$ we have $M'_s = \frac{dM_s}{di} > 0$. Prices are again assumed to be exogenous at some initial level, as in the previous model with exogenous money supply and endogenous money demand.

$$Y = C(Y) + I(Y, i) + G_o \qquad 0 < C_Y = \frac{dC}{dY} < 1 \qquad I_Y = \frac{dI}{dY} > 0 \qquad I' = \frac{dI}{di} < 0 \tag{83}$$

$$\frac{M_s(i)}{p_o} = L_o \qquad M'_s = \frac{dM_s}{di} > 0 \tag{84}$$

Rewriting in an implicit form and solving the matrix equation for the effect of exogenous money demand,

$$Y - C(Y) - I(Y, i) - G_o = 0 \tag{85}$$

$$\frac{M_s(i)}{p_o} - L_o = 0 \tag{86}$$

For the effect of inflation presented as the increase in the price level,

$$\begin{bmatrix} 1 - C_Y - I_Y & -I' \\ 0 & \frac{M'_s}{p_o} \end{bmatrix} \begin{bmatrix} \frac{d\overline{Y}}{dp_o} \\ \frac{d\overline{i}}{dp_o} \end{bmatrix} = \begin{bmatrix} 0 \\ \frac{M_s}{p_o^2} \end{bmatrix} \tag{87}$$

$$|J| = \frac{M'_s(1 - C_Y - I_Y)}{p_o} > 0 \tag{88}$$

$$\frac{d\overline{Y}}{dp_o} = \frac{\begin{vmatrix} 0 & -I' \\ \frac{M_s}{p_o^2} & \frac{M'_s}{p_o} \end{vmatrix}}{|J|} = \frac{I'M_s}{p_o M'_s(1 - C_Y - I_Y)} < 0 \tag{89}$$

$$\frac{d\overline{i}}{dp_o} = \frac{\begin{vmatrix} 1 - C_Y - I_Y & 0 \\ -L_Y & \frac{M_s}{p_o^2} \end{vmatrix}}{|J|} = \frac{M_s}{p_o M'_s} > 0 \tag{90}$$

These results are consistent with the case of exogenous money supply presented above.

$$\begin{bmatrix} 1 - C_Y - I_Y & -I' \\ 0 & \frac{M'_s}{p_o} \end{bmatrix} \begin{bmatrix} \frac{d\overline{Y}}{dL_o} \\ \frac{d\overline{i}}{dL_o} \end{bmatrix} = \begin{bmatrix} 0 \\ 1 \end{bmatrix} \tag{91}$$

$$\frac{d\overline{Y}}{dL_o} = \frac{\begin{vmatrix} 0 & -I' \\ 1 & \frac{M'_s}{p_o} \end{vmatrix}}{|J|} = \frac{I'p_o}{M'_s(1 - C_Y - I_Y)} < 0 \tag{92}$$

$$\frac{d\overline{i}}{dL_o} = \frac{\begin{vmatrix} 1 - C_Y - I_Y & 0 \\ 0 & 1 \end{vmatrix}}{|J|} = \frac{p_o}{M'_s} > 0 \tag{93}$$

Inflationary trends reduce national income and increase the equilibrium interest rate. Money demand has a similar effect. Increased money demand raises the equilibrium interest rate, which leads to a reduction in national income. An expanded version of the model is one where prices are assumed to be endogenous.

$$Y = C(Y, p) + I(Y, i) + G_o \qquad 0 < C_Y < 1 \qquad C_p, I' < 0 \qquad I_Y > 0 \tag{94}$$

$$\frac{M_s(i)}{p} = L_o \qquad M'_s = \frac{dM_s}{di} > 0 \tag{95}$$

$$p = p_o + g(Y) \qquad g_Y > 0 \tag{96}$$

Rewriting the equations

$$Y - C(Y, p) - I(Y, i) - G_o = 0 \tag{97}$$

$$\frac{M_s(i)}{p} - L_o = 0 \tag{98}$$

$$p - p_o - g(Y) = 0 \tag{99}$$

gives rise to the following implicit function theorem.

$$\begin{bmatrix} 1 - C_Y - I_Y & -I' & -C_p \\ 0 & M'_s & -\frac{M_s}{p^2} \\ -g_Y & 0 & 1 \end{bmatrix} \begin{bmatrix} \frac{d\overline{Y}}{dL_o} \\ \frac{d\overline{i}}{dL_o} \\ \frac{d\overline{p}}{dL_o} \end{bmatrix} = \begin{bmatrix} 0 \\ 1 \\ 0 \end{bmatrix} \tag{100}$$

$$|J| = -I'g_Y \frac{M_s}{p^2} + \frac{M'_s}{p}(1 - C_Y - I_Y - g_Y C_p) > 0 \tag{101}$$

$$\frac{d\overline{Y}}{dL_o} = \frac{\begin{vmatrix} 0 & -I' & -C_p \\ 1 & \frac{M'_s}{p} & -\frac{M_s}{p^2} \\ 0 & 0 & 1 \end{vmatrix}}{|J|} = \frac{I'}{-I'g_Y \frac{M_s}{p^2} + \frac{M'_s}{p}(1 - C_Y - I_Y - g_Y C_p)} < 0 \tag{102}$$

$$\frac{d\overline{i}}{dL_o} = \frac{\begin{vmatrix} 1 - C_Y - I_Y & 0 & -C_p \\ 0 & 1 & -\frac{M_s}{p^2} \\ -g_Y & 0 & 1 \end{vmatrix}}{|J|} = \frac{1 - C_Y - I_Y - g_Y C_p}{-I'g_Y \frac{M_s}{p^2} + \frac{M'_s}{p}(1 - C_Y - I_Y - g_Y C_p)} > 0 \tag{103}$$

$$\frac{d\overline{p}}{dL_o} = \frac{\begin{vmatrix} 1 - C_Y - I_Y & -I' & 0 \\ 0 & \frac{M'_s}{p} & 1 \\ -g_Y & 0 & 0 \end{vmatrix}}{|J|} = \frac{g_Y I'}{-I'g_Y \frac{M_s}{p^2} + \frac{M'_s}{p}(1 - C_Y - I_Y - g_Y C_p)} < 0 \tag{104}$$

When demand for money increases, it raises the equilibrium interest rate on the money market. This would be the case of a rightward shift in money demand which, at a particular level of money supply, increases the equilibrium interest rate. A higher interest rate lowers investment and, hence, national income. Due to the fall in aggregate demand, prices also fall. Lower demand for money has the effect of lowering equilibrium prices. Keeping money demand exogenous and adding a price equation, where prices rise with the level of national income, expands the model. It is worth noting that money supply depends positively on the interest rate but negatively on the price level. A higher price level forces the central bank to limit the quantity of money in circulation in order to control inflation.

$$Y = C(Y, p) + I(Y, i) + G_o \qquad 0 < C_Y < 1 \qquad C_p, I' < 0 \qquad I_Y > 0 \tag{105}$$

$$M_s(i, p) = L_o \qquad M_s' = \frac{dM_s}{di} > 0 \qquad M_{sp} = \frac{dM_s}{dp} < 0 \tag{106}$$

$$p = p_o + g(Y) \qquad g_Y > 0 \tag{107}$$

To find the effect of increased money demand on the endogenous variables, we rewrite the equations.

$$Y - C(Y, p) - I(Y, i) - G_o = 0 \tag{108}$$

$$M_s(i, p) - L_o = 0 \tag{109}$$

$$p - p_o - g(Y) = 0 \tag{110}$$

$$\begin{bmatrix} 1 - C_Y - I_Y & -I' & -C_p \\ 0 & M_s' & M_{sp} \\ -g_Y & 0 & 1 \end{bmatrix} \begin{bmatrix} \frac{dY}{dL_o} \\ \frac{di}{dL_o} \\ \frac{dp}{dL_o} \end{bmatrix} = \begin{bmatrix} 0 \\ 1 \\ 0 \end{bmatrix} \tag{111}$$

$$|J| = \underset{+}{I' g_Y M_{sp}} + \underset{+}{M_s' (1 - C_Y - I_Y - g_Y C_p)} > 0 \tag{112}$$

$$\frac{dY}{dL_o} = \frac{\begin{vmatrix} 0 & -I' & -C_p \\ 1 & M_s' & M_{sp} \\ 0 & 0 & 1 \end{vmatrix}}{|J|} = \frac{I'}{\underset{+}{I' g_Y M_{sp}} + \underset{+}{M_s' (1 - C_Y - I_Y - g_Y C_p)}} < 0 \tag{113}$$

$$\frac{di}{dL_o} = \frac{\begin{vmatrix} 1 - C_Y - I_Y & 0 & -C_p \\ 0 & 1 & M_{sp} \\ -g_Y & 0 & 1 \end{vmatrix}}{|J|} = \frac{1 - C_Y - I_Y - g_Y C_p}{\underset{+}{I' g_Y M_{sp}} + \underset{+}{M_s' (1 - C_Y - I_Y - g_Y C_p)}} > 0 \tag{114}$$

$$\frac{dp}{dL_o} = \frac{\begin{vmatrix} 1 - C_Y - I_Y & -I' & 0 \\ 0 & M_s' & 1 \\ -g_Y & 0 & 0 \end{vmatrix}}{|J|} = \frac{g_Y I'}{\underset{+}{I' g_Y M_{sp}} + \underset{+}{M_s' (1 - C_Y - I_Y - g_Y C_p)}} < 0 \tag{115}$$

Higher money demand raises the equilibrium interest rate on the money market. This lowers investment and national income. Due to the fall in aggregate demand, prices also fall. These results confirm what we found previously about the effect of money demand.

## 4. A National Income Model in the Conditions of an Open Economy

The national income model can be expanded by the level of net exports as the difference between exports and imports. This allows for accounting for foreign trade.

$$Y = C(Y, p) + I(Y, i) + G_o + X_o - M(Y, p) \qquad 0 < C_Y, M_Y < 1 \quad C_p, I' < 0 \quad I_Y, M_p > 0 \tag{116}$$

$$\frac{M_{so}}{p} = L(Y, i) \qquad L_Y > 0 \quad L_i < 0 \tag{117}$$

$$p = p_o + g(Y) \qquad g_Y > 0 \tag{118}$$

For the marginal propensity to import $M_Y$, we have $M_Y \in (0, 1)$. In theory, for the three propensities, we have $C_Y + S_Y + M_Y = 1$; that is, the sum of the marginal propensity to consume, to save, and to import equals 1 since this is how income is distributed. Part of the national income goes to consuming domestic goods, part goes to consuming foreign goods, and part is being saved. It is also assumed that imports are positively related to domestic prices; that is, higher domestic prices stimulate the nation to import more from abroad. When domestic prices are considerably high, it is likely that foreign prices would be lower or around the level of domestic prices, which might stimulate consumers to demand more of the foreign goods. Therefore, $M_p > 0$. Given the equilibrium in the capital market, we have $S_Y = I_Y$; that is, if savings equal investment, both represent an equal share of the national income. Note that we also account for the money market. We find the effect of exogenous exports on national income, interest rate, and domestic prices. We also find the comparative static effect of an increase in government spending and compare the results with those of a closed economy. Rewriting the system in an implicit form,

$$Y - C(Y, p) - I(Y, i) - G_o - X_o + M(Y, p) = 0 \tag{119}$$

$$L(Y, i) - \frac{M_{so}}{p} = 0 \tag{120}$$

$$p - p_o - g(Y) = 0 \tag{121}$$

$$\begin{bmatrix} 1 - C_Y - I_Y + M_Y & -I' & -C_p + M_p \\ L_Y & L_i & \frac{M_{so}}{p^2} \\ -g_Y & 0 & 1 \end{bmatrix} \begin{bmatrix} \frac{d\overline{Y}}{dX_o} \\ \frac{d\overline{i}}{dX_o} \\ \frac{d\overline{p}}{dX_o} \end{bmatrix} = \begin{bmatrix} 1 \\ 0 \\ 0 \end{bmatrix} \tag{122}$$

$$|J| = I' \left( \underbrace{L_Y + g_Y \frac{M_{so}}{p^2}}_{-} \right) + L_i \left[ \underbrace{1 - C_Y - I_Y + M_Y - g_Y(C_p - M_p)}_{-} \right] < 0 \tag{123}$$

Since $C_Y$ and $I_Y$ are marginal propensities, respectively, to consume and to save, where investment equals savings in equilibrium, their sum is less than 1. Hence, the second parenthesized term is strictly positive, which gives a negative Jacobian.

$$\frac{d\overline{Y}}{dX_o} = \frac{\begin{vmatrix} 1 & -I' & -C_p + M_p \\ 0 & L_i & \frac{M_{so}}{p^2} \\ 0 & 0 & 1 \end{vmatrix}}{|J|} = \frac{L_i}{I' \left( L_Y + g_Y \frac{M_{so}}{p^2} \right) + L_i \left[ 1 - C_Y - I_Y + M_Y - g_Y(C_p - M_p) \right]} > 0 \tag{124}$$

$$\frac{d\overline{i}}{dX_o} = \frac{\begin{vmatrix} 1 - C_Y - I_Y + M_Y & 1 & -C_p + M_p \\ L_Y & 0 & \frac{M_{so}}{p^2} \\ -g_Y & 0 & 1 \end{vmatrix}}{|J|} = -\frac{L_Y + g_Y \frac{M_{so}}{p^2}}{I' \left( L_Y + g_Y \frac{M_{so}}{p^2} \right) + L_i \left[ 1 - C_Y - I_Y + M_Y - g_Y(C_p - M_p) \right]} > 0 \tag{125}$$

$$\frac{d\overline{p}}{dX_o} = \frac{\begin{vmatrix} 1 - C_Y - I_Y + M_Y & -I' & 1 \\ L_Y & L_i & 0 \\ -g_Y & 0 & 0 \end{vmatrix}}{|J|} = \frac{L_i g_Y}{I' \left( L_Y + g_Y \frac{M_{so}}{p^2} \right) + L_i \left[ 1 - C_Y - I_Y + M_Y - g_Y(C_p - M_p) \right]} > 0 \tag{126}$$

The effect of exports is similar to that of government spending. Exports stimulate national income, which further increases the interest rate and domestic prices. A country that exports heavily eventually achieves a higher standard of living. Countries which

pursue export-led growth eventually see their prices rising and their currency appreciating against foreign currencies. Such examples are Japan and South Korea in the past and, presently, China. To find the effect of government spending, we find the three comparative-static derivatives; that is,

$$
\begin{bmatrix} 1 - C_Y - I_Y + M_Y & -I' & -C_p + M_p \\ L_Y & L_i & \frac{M_{so}}{p^2} \\ -g_Y & 0 & 1 \end{bmatrix} \begin{bmatrix} \frac{dY}{dG_o} \\ \frac{di}{dG_o} \\ \frac{dp}{dG_o} \end{bmatrix} = \begin{bmatrix} 1 \\ 0 \\ 0 \end{bmatrix} \tag{127}
$$

$$
|J| = I' \left( L_Y + g_Y \frac{M_{so}}{p^2} \right) + L_i \left[ 1 - C_Y - I_Y + M_Y - g_Y (C_p - M_p) \right] < 0 \tag{128}
$$

$$
\frac{d\overline{Y}}{dG_o} = \frac{\begin{vmatrix} 1 & -I' & -C_p + M_p \\ 0 & L_i & \frac{M_{so}}{p^2} \\ 0 & 0 & 1 \end{vmatrix}}{|J|} = \frac{L_i}{I' \left( L_Y + g_Y \frac{M_{so}}{p^2} \right) + L_i \left[ 1 - C_Y - I_Y + M_Y - g_Y (C_p - M_p) \right]} > 0 \tag{129}
$$

$$
\frac{d\overline{i}}{dG_o} = \frac{\begin{vmatrix} 1 - C_Y - I_Y + M_Y & 1 & -C_p + M_p \\ L_Y & 0 & \frac{M_{so}}{p^2} \\ -g_Y & 0 & 1 \end{vmatrix}}{|J|} = -\frac{L_Y + g_Y \frac{M_{so}}{p^2}}{I' \left( L_Y + g_Y \frac{M_{so}}{p^2} \right) + L_i \left[ 1 - C_Y - I_Y + M_Y - g_Y (C_p - M_p) \right]} > 0 \tag{130}
$$

$$
\frac{d\overline{p}}{dG_o} = \frac{\begin{vmatrix} 1 - C_Y - I_Y + M_Y & -I' & 1 \\ L_Y & L_i & 0 \\ -g_Y & 0 & 0 \end{vmatrix}}{|J|} = \frac{L_i g_Y}{I' \left( L_Y + g_Y \frac{M_{so}}{p^2} \right) + L_i \left[ 1 - C_Y - I_Y + M_Y - g_Y (C_p - M_p) \right]} > 0 \tag{131}
$$

The comparative static derivatives for government spending are the same as those for exports in the open economy model. The exports multiplier $\frac{d\overline{Y}}{dX_o}$ is equal to the government expenditure multiplier $\frac{d\overline{Y}}{dG_o}$. Furthermore, the government expenditure multiplier in a closed economy is the same as that in an open economy, i.e., if foreign trade is ignored. In the absence of exports and imports for the country the terms $M_Y$ (the marginal propensity to import) and $M_p$ (dependence of imports on domestic prices) are dropped from the system.

$$
\frac{d\overline{Y}}{dG_o} = \frac{1}{\frac{I'}{L_i} \left( L_Y + g_Y \frac{M_{so}}{p^2} \right) + 1 - C_Y - I_Y - g_Y C_p} > 0, \tag{132}
$$

gives the government expenditure multiplier in a closed economy.

$$
\frac{d\overline{Y}}{dG_o} = \frac{1}{\frac{I'}{L_i} \left( L_Y + g_Y \frac{M_{so}}{p^2} \right) + 1 - C_Y - I_Y + M_Y - g_Y C_p + g_Y M_p} > 0, \tag{133}
$$

gives the government expenditure multiplier in an open economy. Since the terms $M_Y$ and $g_Y M_p$ are positive and are added to the denominator, the government expenditure multiplier is smaller for an open economy than it is for a closed one. The same applies to the equilibrium interest rate and prices, which seem to rise more slowly in an open economy. For the equilibrium interest rate and the average price level in a closed economy, we previously found the following:

$$
\frac{d\overline{i}}{dG_o} = -\frac{L_Y + g_Y \frac{M_{so}}{p^2}}{I' (L_Y + g_Y \frac{M_{so}}{p^2}) + L_i (1 - C_Y - I_Y - g_Y C_p)} > 0,
$$

from Equation (46)

$$\frac{d\overline{p}}{dG_o} = \frac{L_i g_Y}{I'(L_Y + g_Y \frac{M_{so}}{p^2}) + L_i(1 - C_Y - I_Y - g_Y C_p)} > 0,$$

and Equation (47).

Alternatively, we can assume that imports are exogenous while exports depend positively on national income and negatively on the price level. If domestic prices are higher and there is high demand for local goods, then exporters are less likely to export their goods abroad. The model thus becomes as follows:

$$Y = C(Y,p) + I(Y,i) + G_o + X(Y,p) - M_o \qquad 0 < C_Y < 1 \qquad C_p, I', X_p < 0 \qquad I_Y, X_Y > 0 \tag{134}$$

$$\frac{M_{so}}{p} = L(Y,i) \qquad L_Y > 0 \qquad L_i < 0 \tag{135}$$

$$p = p_o + g(Y) \qquad g_Y > 0 \tag{136}$$

The effect of imports on the endogenous variables can be found as follows:

$$Y - C(Y,p) - I(Y,i) - G_o - X(Y,p) + M_o = 0 \tag{137}$$

$$L(Y,i) - \frac{M_{so}}{p} = 0 \tag{138}$$

$$p - p_o - g(Y) = 0 \tag{139}$$

$$\begin{bmatrix} 1 - C_Y - I_Y - X_Y & -I' & -C_p - X_p \\ L_Y & L_i & \frac{M_{so}}{p^2} \\ -g_Y & 0 & 1 \end{bmatrix} \begin{bmatrix} \frac{dY}{dM_o} \\ \frac{di}{dM_o} \\ \frac{dp}{dM_o} \end{bmatrix} = \begin{bmatrix} -1 \\ 0 \\ 0 \end{bmatrix} \tag{140}$$

$$|J| = I'\left(L_Y + g_Y \frac{M_{so}}{p^2}\right) + L_i[1 - C_Y - I_Y - X_Y - g_Y(C_p + X_p)] < 0 \tag{141}$$

If we assume that in equilibrium investment should grow at the same rate as savings and exports at the rate of imports, the second parenthesized term is positive, which gives a negative Jacobian.

$$\frac{dY}{dM_o} = \frac{\begin{vmatrix} -1 & -I' & -C_p - X_p \\ 0 & L_i & \frac{M_{so}}{p^2} \\ 0 & 0 & 1 \end{vmatrix}}{|J|} = -\frac{L_i}{I'\left(L_Y + g_Y \frac{M_{so}}{p^2}\right) + L_i[1 - C_Y - I_Y - X_Y - g_Y(C_p + X_p)]} < 0 \tag{142}$$

$$\frac{di}{dM_o} = \frac{\begin{vmatrix} 1 - C_Y - I_Y - X_Y & -1 & -C_p - X_p \\ L_Y & 0 & \frac{M_{so}}{p^2} \\ -g_Y & 0 & 1 \end{vmatrix}}{|J|} = \frac{L_Y + g_Y \frac{M_{so}}{p^2}}{I'\left(L_Y + g_Y \frac{M_{so}}{p^2}\right) + L_i[1 - C_Y - I_Y - X_Y - g_Y(C_p + X_p)]} < 0 \tag{143}$$

$$\frac{dp}{dM_o} = \frac{\begin{vmatrix} 1 - C_Y - I_Y - X_Y & -I' & -1 \\ L_Y & L_i & 0 \\ -g_Y & 0 & 0 \end{vmatrix}}{|J|} = -\frac{L_i g_Y}{I'\left(L_Y + g_Y \frac{M_{so}}{p^2}\right) + L_i[1 - C_Y - I_Y - X_Y - g_Y(C_p + X_p)]} < 0 \tag{144}$$

We see that the effect of imports is just opposite to that of exports. Imports reduce national income, thus bringing about lower interest rates and domestic prices. A country

that becomes indebted and has imports that grow way above its exports would eventually experience a declining living standard and falling prices. This is consistent with the price–specie flow mechanism in international trade by which a negative trade balance can be cleared automatically in the absence of trade barriers. The interplay of exports and imports and their combined effect on national wealth, interest rate, and the average price level confirm the price–specie flow, namely, that heavy exporters realize a positive trade balance and become wealthy, which increases their standard of living and the price level. This allows them to import more, which eventually turns them into heavy importers. With the passage of time, these countries see their prices and interest rates falling and experience a negative trade balance which stimulates them to export again. The mechanism resembles a pendulum, where a country involved in foreign trade would gravitate around the zero-trade balance.

We previously discussed the effect of exogenous exports on the average price level and the exchange rate. The open economy model can further be expanded to account for the exchange rate $e$. Consistent with economic theory, a higher exchange rate stimulates imports but discourages exports; that is, $X_e < 0$ and $M_e > 0$. Furthermore, it is expected that money supply depends positively on the exchange rate, since a higher exchange rate is associated with a higher interest rate. To balance the interest rate, the central bank will increase the quantity of money in circulation when the exchange rate grows. This time, we find the effect of government spending and the initial price level on the endogenous variables, national income $Y$, the exchange rate $e$, and the price level $p$.

$$Y = C(Y,p) + I(Y) + G_o + X(e) - M(Y,e) \qquad 0 < C_Y, M_Y < 1 \qquad C_p, X_e < 0 \qquad I_Y, M_e > 0 \tag{145}$$

$$M_s(e,p) = L(Y) \qquad M_{se}, L_Y > 0 \qquad M_{sp}, L_i < 0 \tag{146}$$

$$p = p_o + g(Y) \qquad g_Y > 0 \tag{147}$$

Solving the system,

$$Y - C(Y,p) - I(Y) - G_o - X(e) + M(Y,e) = 0 \tag{148}$$

$$M_s(e,p) - L(Y) = 0 \tag{149}$$

$$p - p_o - g(Y) = 0 \tag{150}$$

$$|J| = g_Y \underbrace{[C_p M_{se} - M_{sp}(X_e - M_e)]}_{-} - M_{se} \underbrace{(1 - C_Y - I_Y + M_Y)}_{+} + L_Y \underbrace{(X_e - M_e)}_{-} < 0 \tag{151}$$

To find the effect of exogenous government spending on $Y$, $p$, and $e$:

$$\begin{bmatrix} 1 - C_Y - I_Y + M_Y & -C_p & -X_e + M_e \\ -L_Y & M_{sp} & M_{se} \\ -g_Y & 1 & 0 \end{bmatrix} \begin{bmatrix} \frac{dY}{dG_o} \\ \frac{dp}{dG_o} \\ \frac{de}{dG_o} \end{bmatrix} = \begin{bmatrix} 1 \\ 0 \\ 0 \end{bmatrix} \tag{152}$$

$$|J| = g_Y \underbrace{[C_p M_{se} - M_{sp}(X_e - M_e)]}_{-} - M_{se} \underbrace{(1 - C_Y - I_Y + M_Y)}_{+} + L_Y \underbrace{(X_e - M_e)}_{-} < 0 \tag{153}$$

$$\frac{dY}{dG_o} = \frac{\begin{vmatrix} 1 & -C_p & -X_e + M_e \\ 0 & M_{sp} & M_{se} \\ 0 & 1 & 0 \end{vmatrix}}{|J|} = -\frac{M_{se}}{g_Y \underbrace{[C_p M_{se} - M_{sp}(X_e - M_e)]}_{-} - M_{se} \underbrace{(1 - C_Y - I_Y + M_Y)}_{+} + L_Y \underbrace{(X_e - M_e)}_{-}} > 0 \tag{154}$$

$$\frac{dp}{dG_o} = \frac{\begin{vmatrix} 1 - C_Y - I_Y + M_Y & 1 & -X_e + M_e \\ -L_Y & 0 & M_{se} \\ -g_Y & 0 & 0 \end{vmatrix}}{|J|} = -\frac{g_Y M_{se}}{g_Y \underbrace{[C_p M_{se} - M_{sp}(X_e - M_e)]}_{-} - M_{se} \underbrace{(1 - C_Y - I_Y + M_Y)}_{+} + L_Y \underbrace{(X_e - M_e)}_{-}} > 0 \tag{155}$$

$$\frac{de}{dG_o} = \frac{\begin{vmatrix} 1 - C_Y - I_Y + M_Y & -C_p & 1 \\ -L_Y & -M_{sp} & 0 \\ -g_Y & 1 & 0 \end{vmatrix}}{|J|} = -\frac{L_Y + g_Y M_{se}}{g_Y[C_p M_{se} - M_{sp}(X_e - M_e)] - M_{se}(1 - C_Y - I_Y + M_Y) + L_Y(X_e - M_e)} > 0 \quad (156)$$

Increased government spending stimulates national income through the multiplier process. This increases aggregate demand, which causes prices to rise. These results are consistent with what we found about the macroeconomics effects of government spending in the conditions of a closed economy. The effect on the exchange rate turns out to also be favorable. Through national income, the government spending increases money demand and the average price level, which ultimately leads to an increase in the exchange rate. For the effect of exogenous initial prices $p_o$, we have

$$\begin{bmatrix} 1 - C_Y - I_Y + M_Y & -C_p & -X_e + M_e \\ -L_Y & M_{sp} & M_{se} \\ -g_Y & 1 & 0 \end{bmatrix} \begin{bmatrix} \frac{dY}{dp_o} \\ \frac{dp}{dp_o} \\ \frac{de}{dp_o} \end{bmatrix} = \begin{bmatrix} 0 \\ 0 \\ 1 \end{bmatrix} \quad (157)$$

$$|J| = g_Y[C_p M_{se} - M_{sp}(X_e - M_e)] - M_{se}(1 - C_Y - I_Y + M_Y) + L_Y(X_e - M_e) < 0 \quad (158)$$

$$\frac{dY}{dp_o} = \frac{\begin{vmatrix} 0 & -C_p & -X_e + M_e \\ 0 & M_{sp} & M_{se} \\ 1 & 1 & 0 \end{vmatrix}}{|J|} = -\frac{C_p M_{se} - M_{sp}(X_e - M_e)}{g_Y[C_p M_{se} - M_{sp}(X_e - M_e)] - M_{se}(1 - C_Y - I_Y + M_Y) + L_Y(X_e - M_e)} < 0 \quad (159)$$

$$\frac{dp}{dp_o} = \frac{\begin{vmatrix} 1 - C_Y - I_Y + M_Y & 0 & -X_e + M_e \\ -L_Y & 0 & M_{se} \\ -g_Y & 1 & 0 \end{vmatrix}}{|J|} = -\frac{(1 - C_Y - I_Y + M_Y)M_{se} - L_Y(X_e - M_e)}{g_Y[C_p M_{se} - M_{sp}(X_e - M_e)] - M_{se}(1 - C_Y - I_Y + M_Y) + L_Y(X_e - M_e)} > 0 \quad (160)$$

$$\frac{de}{dp_o} = \frac{\begin{vmatrix} 1 - C_Y - I_Y + M_Y & -C_p & 0 \\ -L_Y & -M_{sp} & 0 \\ -g_Y & 1 & 1 \end{vmatrix}}{|J|} = -\frac{(1 - C_Y - I_Y + M_Y)M_{sp} + L_Y C_p}{g_Y[C_p M_{se} - M_{sp}(X_e - M_e)] - M_{se}(1 - C_Y - I_Y + M_Y) + L_Y(X_e - M_e)} < 0 \quad (161)$$

Inflation reduces national income and results in higher prices. A higher initial price level fosters future inflation. Due to the reduction in national income, inflation also eventually reduces the exchange rate, which is why the national currency depreciates against other currencies. These results confirm what we found about the effect of inflation previously.

## 5. Conclusions

A brief microeconomic analysis demonstrates that demand shifters such as consumer income (for normal goods) and advertising by firms increase market demand and thus raise equilibrium price in a partial market equilibrium model. Supply shifters such as the tax level or generally the price of inputs reduce supply, which increases equilibrium price. The effect of technology is the opposite, reducing the price at a particular level of market demand.

The macroeconomic analysis confirms the main findings of Keynesian theory. We obtain the standard government expenditure multiplier $\frac{d\overline{Y}}{dG_o} = \frac{1}{1 - C_Y} > 0$ to be positive, but we expand the multiplier to account for taxes, the average price level, the money market, and foreign trade. In all cases, the government expenditure multiplier is positive. The effect of government spending on prices is also positive, which is consistent with Keynesian theory. This effect is confirmed both in the conditions of a closed and an open economy. While government spending can be a trigger for inflation, taxes have the opposite effect.

They can prevent inflation and offset the effect of government spending, as predicted by John Maynard Keynes.

Again, consistent with Keynesian theory, money supply increases national income. The monetarist school supports the view that money supply increases only nominal national income without affecting real one. In other words, money supply would have an inflationary effect on the economy. By reducing the interest rate, money supply stimulates investment which increases aggregate demand and prices further. This confirms the monetarist view of the inflationary effects of money supply. The effect of money demand is just the opposite. Increased money demand raises the equilibrium interest rate, which hampers aggregate investment and lowers national income. In this way, money demand lowers the price level.

Inflation or the increase in the initial price level affects national income adversely. Increased prices hamper aggregate consumption, which leads to lower national income. Inflation also raises the equilibrium interest rate, which hinders investment. In the open economy model, inflation reduces the exchange rate, that is, the national currency depreciates against other currencies. As predicted by Keynesian theory, the exports multiplier is positive; that is, exports have a positive effect on national income. Exports also raise the equilibrium interest rate and the average price level. The effect of exogenous imports on the economy is the opposite. Imports lower national income, the equilibrium interest rate, and the overall price level. Our analysis confirms all major findings of Keynesian theory, although proving it was not our initial intention.

**Author Contributions:** Conceptualization, methodology, formal analysis, writing—original draft and preparation, T.P.T.; review and editing, B.M. All authors have read and agreed to the published version of the manuscript.

**Funding:** This research received no funding.

**Institutional Review Board Statement:** Not applicable.

**Informed Consent Statement:** Not applicable.

**Data Availability Statement:** All data are contained withing the article.

**Conflicts of Interest:** The authors declare no conflict of interest.

## Notes

1. We do not aim a full review of all theories of inflation in this study.

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
