# Peer review of "Factors Determining the Average Price Level: A Combined Microeconomic and Macroeconomic Approach"

_economies, doi:10.3390/economies12050121_

Round 1
Reviewer 1 Report
Comments and Suggestions for Authors
• Empirical findings emphasize the Macroeconomic approach in determining the average price level. The microeconomic approach also needs to be given proportion in the analysis of study results and the relationship between the two approaches in developing a model for determining the average price level. Therefore, the microeconomic approach is an important part that needs to be explained in the background, as how the combination of the two becomes a novelty for the study.
• The formation of a model determining the average price level from both a microeconomic and a macroeconomic perspective necessitates a robust literature review, encompassing both theoretical and empirical aspects. Hence, this study should clearly articulate the novelty and contribution of the research, which is underpinned by a comprehensive understanding of the existing body of knowledge.
• The model for determining the average price level, which combines Microeconomic and Macroeconomic approaches, would be significantly enhanced if it were substantiated by empirical evidence. This would not only validate the model but also allow for further discussion of the results, thereby increasing the research's applicability and relevance.
• Conclusions can be presented more concisely by adding the limitations of the model formed and opportunities that can be revealed for future model development
Author Response
Thank you for your suggestions. The goal of the paper is rather modest. It is a theoretical, quantitative paper rather than an empirical one. The uniqueness of the paper is in the development of the mathematical model. We have not seen such a presentation in the literature and what looks like obvious results may not be so obvious. Upon your advice we have shortened the conclusion as much as possible. It incorporates all essential findings and mathematical results.
Reviewer 2 Report
Comments and Suggestions for Authors
economies-2981420
Factors Determining the Average Price Level: A Combined Microeconomic and Macroeconomic Approach
The paper is an interesting exercise in Comparative Statics. The authors, applying a basic framework within the national income model reveals that government spending boosts national income, the equilibrium interest rate, and the price level. Conversely, incorporating taxes into the model shows that they exert the opposite influence on the macroeconomic variables, dampening inflationary pressure on prices. Furthermore, we observe that an increase in the money supply raises national income and the price level while decreasing the equilibrium interest rate. Conversely, an increase in money demand has the opposite effect, reducing national income and the price level while elevating the interest rate. Finally, they infer that while they may have differing approaches, Keynesian and monetarist views on inflation are not necessarily mutually exclusive or opposing.
1). The authors should justify and explain their findings. Also, they have to clearly state what is the added value of their work to literature. Apparently, all this is well known.
2). The analysis presented in this paper is not comprehensive. Keynesian and monetarist views on inflation are not necessarily opposing but often differ in their emphasis and proposed solutions. Yet, differences in intentions and ideology contribute to the varying perspectives on inflation between Keynesian and monetarist economists. Keynesian economics emphasizes the importance of government intervention in managing the economy, particularly during periods of economic downturns such as recessions. On the other hand, monetarist economics places greater emphasis on the role of the money supply and the central bank in influencing economic outcomes. Monetarists advocate for a rules-based approach to monetary policy, emphasizing the importance of controlling the growth rate of the money supply to maintain stable prices and curb inflation. These differences in intentions and ideology lead to contrasting policy prescriptions and views on the appropriate role of government intervention in managing the economy. While Keynesians may be more willing to tolerate inflation to achieve broader economic objectives such as full employment, monetarists prioritize price stability and advocate for tighter control over the money supply to curb inflationary pressures.
However, there is another school, the Marxist school of economics that argue that inflation arises from contradictions inherent in capitalist production, particularly the tendency for the rate of profit to decline over time, and view inflation as a symptom of deeper structural issues within capitalist economies rather than as a standalone problem. According to that school of economic thought, inflation can result from various factors, including the concentration and centralization of capital, unequal distribution of wealth, and the exploitation of labor by capital.
3). An exhaustive mathematical presentation is not necessary. It resembles course notes.
4). The Conclusion section is too long and should be shortened.
5). Finally, what is the outcome of the paper? The main conclusion?
6). References should be updated.
Author Response
We appreciate the comment on the policy implications of Keynesian and monetarist views of inflation. We have added a sentence on the different prescriptions and effects of both schools of economic thought. We find the comment substantive. We do not aim for a comprehensive study of all theories of inflation, and we mention this in the paper. We are not overly ambitious on covering all theories, including Marxism. A weakness of the Marxist school is that it provides thorough criticism to capitalist societies without suggesting an alternative social model. This is evident from the failure of the Marxist experiment in Eastern Europe. Being East European, the authors are particularly sensitive to the topic.
We have reduced the conclusion as much as possible. The conclusion summarizes the essential findings of the paper. We have added 4 recent references in the context of the structural theory of inflation and the rational expectations theory. Thank you for your constructive comments and suggestions.
Reviewer 3 Report
Comments and Suggestions for Authors
Firstly, mixing amcro and micro perspective might not be a good idea.
But OK, if you decide to follow the mixing path, please provide some empirical supports for your toughts.
This paper is mathematicaly and theorethically consistent. However, not much new was provided in the paper - your mathematical illuastration are a bit selfevident.
To improve the manucript, please extend your manucript with a new section empirical illustration.
Author Response
Thank you for your comments. What looks like obvious results may not be so obvious. Our paper is unique in the premises of our model. We have not seen such a presentation of mathematical relationships and assumptions in other such papers. There are a few empirical papers devoted to inflation but very few theoretical. Ours is a purely theoretical, mathematical paper and not an empirical one. It is quite long, as it is. Extending it with an empirical section will make it unnecessarily long and will change its focus.
We are glad you find our analysis mathematically consistent and do not have criticism to our equations which are essentially the substance of the paper.
Round 2
Reviewer 1 Report
Comments and Suggestions for Authors
· The background focuses more on the debate on the issue of inflation from macroeconomic approaches from several schools, both classical and modern. We also want to know the microeconomic school that debates the origins of inflation. • The paper is limited to establishing a mathematical model of the determinants of inflation using a combination of Microeconomic and Macroeconomic approaches. However, what is more essential for us to understand is the theoretical support of the model developed and empirical verification. Besides that, we must know the position of the mathematical model being created compared to previous models that have been tested empirically
Author Response
Thank you for your remarks. We have discussed at length the theoretical underpinnings of inflation from the perspective of the Keynesian and monetarist school. We find confirmation of Keynesian main findings. This is essentially the model which serves as a benchmark to our mathematical derivations. We quote Keynes (1936) and Friedman (1970). These are the two models we compare our results to. Do you have any particular models in mind which we could include in our paper? If you do, can you please refer us to those so we can incorporate them in the paper? Or you have particular sources in mind which you think we should refer to? If there are any particular models or sources you recommend, please let us know.
The microeconomic part is merely an introduction into the macroeconomic section. The paper is essentially a macroeconomic analysis. We have not seen any such coverage and presentation of the relationships using these specific equations and assumptions. The paper is unique in its equations and mathematical approach to the treatment of the macroeconomic variables including an open economy model with exports and imports considered. It is thereof a theoretical paper on inflation, not an empirical one. Writing it empirically would produce a different and an entirely new paper.
Reviewer 2 Report
Comments and Suggestions for Authors
Thank you for revising
Author Response
Thank you for your support.
Reviewer 3 Report
Comments and Suggestions for Authors
I support this paper in the for as is.
Author Response
Thank you for your support.